# Transcriptome-Wide Identification of Neuropeptides and Neuropeptide Receptors in the Twenty-Eight-Spotted Ladybird *Henosepilachna vigintioctopunctata*

**DOI:** 10.3390/insects16060624

**Published:** 2025-06-13

**Authors:** Quanxing Lei, Ziming Wang, Shuangyan Yao, Aili Lin, Yunhui Zhang, Chengxian Sun, Xiaoguang Liu, Mengfang Du, Xiaoming Liu, Shiheng An

**Affiliations:** 1Henan International Laboratory for Green Pest Control, College of Plant Protection, Henan Agricultural University, Zhengzhou 450046, China; leiquanxing526@163.com (Q.L.); zimingwang2024@163.com (Z.W.); yaoshuangyan@outlook.com (S.Y.); 13403721591@163.com (Y.Z.); wsyenan@163.com (C.S.); xgliu2000@aliyun.com (X.L.); dumengfang@163.com (M.D.); anshiheng@aliyun.com (S.A.); 2Henan International Joint Laboratory of Taxonomy and Systematic Evolution of Insecta, Henan Institute of Science and Technology, Xinxiang 453003, China; linaili2023@126.com

**Keywords:** neuropeptide, G protein-coupled receptor, *Henosepilachna vigintioctopunctata*

## Abstract

The 28-spotted ladybird beetle (*Henosepilachna vigintioctopunctata*) is a major pest that damages crops in the potato/tomato family. Neuropeptides and their receptors might help control this pest. By analyzing the beetle’s central nervous system, we identified 58 neuropeptide genes and 31 receptor genes linked to biological processes. The neuropeptide genes are active in the insect’s brain, ventral nerve cord, and gut. This research provides a foundation for developing eco-friendly pest control methods targeting these neuropeptide systems, which could reduce crop damage.

## 1. Introduction

Neuropeptides are classic signaling molecules produced and released by most major types of neurons that are mainly located in the central nervous system, including the brain and ventral nerve cord (VNC) [1]. They are small proteins with generally several to tens of amino acid residues, are one of the structurally most diverse signaling molecules, and are the most diverse group of signaling molecules in multicellular organisms [2,3]. Neuropeptides act via their respective neuropeptide receptors, most of which belong to G protein-coupled receptors [4,5]. Some neuropeptide receptors are not GPCRs, such as the insulin receptor family, which includes three type-II receptor tyrosine kinases (RTKs) that each have one transmembrane domain [6]. It has been widely reported that neuropeptides and their receptors activate the essential signaling pathways that regulate physiological processes such as growth, development, behavior, reproduction, metabolism, and many other physiological processes in insects [1,2].

For example, pheromone biosynthesis-activating neuropeptide (PBAN) activates the synthesis of sex pheromone in Lepidoptera [7,8]. Neuropeptide F (NPF), like neuropeptide Y (NPY) in vertebrates, regulates feeding and metabolism [9]. Insulin-like peptides are known to regulate growth and development, reproduction, stress resistance, and lifespan [10]. The neuropeptide CCHamide (CCH) regulates feeding intake, sensory perception, and olfactory behavior [11,12]. Neuropeptide Bursicon regulates cuticle metabolism of insects and the transition from summer-form to winter-form of *Cacopsylla chinensis* [13,14]. Thus, neuropeptides and their receptors could be developed as potential insecticides or targets for a novel generation of pesticides. Therefore, identification and functional characterization of neuropeptides and their receptors from insect pests would enhance our basic understanding of neuropeptide-related signal transduction and provide important molecular insights for pest management. Up to now, neuropeptides and their receptors have been studied in some species of pests, such as *Nilaparvata lugens* [15], *Aphis craccivora* [16], *Phauda flammans* [17], *Grapholita molesta* [18], and *Eurygaster integriceps* [19]. Although a few studies on neuropeptides and their receptors have been reported in Coleoptera, such as *Tribolium castaneum* [20], *Tenebrio molitor Zophobas atratus* [21], and *Coccinella septempunctata* [22], no such information is available for *Henosepilachna vigintioctopunctata.*

In this study, we conducted high-throughput sequencing of the central nervous system (CNS), identified members of the neuropeptides and neuropeptide receptors of *H. vigintioctopunctata,* and compared them with those reported neuropeptides and neuropeptide receptors of other species. We also evaluated the expression level of 58 neuropeptides in different larval tissues. Our results could provide useful information on neuropeptides and their receptors and a theoretical basis for their functional analysis.

## 2. Materials and Methods

### 2.1. Insect Rearing and Sample Collection

A *H. vigintioctopunctata* colony was reared in the laboratory (Lab of Insect Physiology and Biochemistry in Henan Agricultural University, Zhengzhou, China) for six generations on the leaves of *Solanum nigrum*. They were kept at 26 ± 1 °C, 70 ± 10% relative humidity (RH), under a 14:10 h light: dark photoperiodic regime.

For transcriptome analysis, whole CNS (n = 150 pooled specimens) (Figure 1A) were micro-dissected from second- and third-instar larvae in cold 10 mM phosphate-buffered saline (PBS, pH 7.4). For spatial expression analysis, samples of the brain (Br, n = 150), ventral nerve cord (VNC, n = 150), gut (whole gut including foregut, midgut and hindgut, n = 20), and Malpighian tubule (Mt, n = 30) were dissected from 2-day-old second-instar larvae. All the tissues were immediately pooled in RNase-free tubes and used to extract total RNA.

### 2.2. RNA Extraction and Transcriptome Sequencing

Total RNA was isolated from CNS samples using RNAiso Plus reagent (Takara Bio, Kusatsu, Japan). RNA quality assessment, cDNA library preparation, and transcriptome sequencing were performed by Majorbio Bio-pharm Biotechnology Co., Ltd. (Shanghai, China) on the Illumina HiSeq platform, generating 150 bp paired-end reads. Raw sequencing data were subjected to quality filtering to obtain clean reads using fastp (default parameters), which were subsequently de novo assembled into unigenes using Trinity v2.8.5 (k-mer = 31, min contig length = 200 bp) [23]. Assembly completeness was evaluated with BUSCO v3.0.2 [24] against the insecta_odb9 dataset [24]. Functional annotation was conducted through six databases: Pfam (HMMER3 v3.1b2, default parameters); KEGG (KOBAS v2.1.1, default parameters); EggNOG (DIAMOND v0.9.24, E-value ≤ 1 × 10^5^); Nr (DIAMOND v0.9.24, E-value ≤ 1 × 10^5^); GO (BLAST2GO v2.5.0, default parameters); Swiss-Prot (DIAMOND v0.9.24, E-value ≤ 1 × 10^5^).

### 2.3. Exploration of the Neuropeptides and Their Putative G Protein-Coupled Receptors

Based on known sequences from *Tribolium castaneum* [20,25], *Harmonia axyridis* [26], *Coccinella septempunctata* [22,26], *Drosophila melanogaster* [27], and *Bombyx mori* [28,29,30], the neuropeptide precursor and GPCRs genes of *H. vigintioctopunctata* were identified by tBLASTn analysis (E-value cutoff ≤ 1 × 10⁻⁵ and length filtering: precursor sequences ≥ 30 amino acids) [31].

The neuropeptide precursors of *H. vigintioctopunctata* were classified into distinct families using NeuroPep (version 1.0) [32]. For neuropeptide precursor characterization, N-terminal signal peptides were predicted with the SignalP 5.0 [33], and putative cleavage sites (RR/RK/KK/KR/R/K) were identified based on criteria established in prior research [34] or analyzed through NeuroPred [35]. Multiple sequence alignments were conducted using Clustal Omega (version 1.2.2) [36]. For GPCR analysis, transmembrane domains (TMDs) were predicted via the TMHMM 2.0 [37], and conserved domains were annotated by querying the Conserved Domain Database [38].

### 2.4. Phylogenetic Analysis

Phylogenetic trees of *H. vigintioctopunctata* neuropeptide precursors and neuropeptide GPCRs were reconstructed using sequences from well-characterized species, including *B. mori* [30], *T. castaneum* [25], *Harmonia axyridis* [26] and other species. Amino acid sequences of neuropeptide precursors and GPCRs were retrieved from the NCBI database. All amino acid sequences were aligned using MEGA version 11.0 [39]. The neighbor-joining phylogeny was constructed in MEGA version 11.0 with a p-distance model. Branch robustness was evaluated through 1000 bootstrap replicates, and missing data/gaps were treated via pairwise deletion. The metabotropic glutamate receptor of *Neocloeon triangulifer* (NtrGluR, XP_059491291) was designated as the outgroup for GPCR phylogeny. Final dendrograms were annotated and color-coded using FigTree1.4.4 (http://tree.bio.ed.ac.uk) (accessed on 6 March 2025).

### 2.5. Spatial Expression

Total RNA was extracted from pre-processed tissues using the method described in the preceding section. First-strand cDNA synthesis was carried out with the PrimeScript RT Reagent Kit with gDNA Eraser (Takara Bio, Japan). Quantitative Real-Time PCR (qRT-PCR) reactions were conducted in 20 μL volumes using TB Green™ Premix Ex Taq™ (Takara Bio, Japan) on a CFX Connect Real-Time PCR Detection System (Bio-Rad, Hercules, CA, USA), with the following thermal cycling parameters: initial denaturation at 95 °C for 30 s; 40 cycles of 95 °C for 5 s, 60 °C for 30 s; and a melt curve analysis from 65 °C to 95 °C with 0.5 °C increments per 5 s. Two ribosomal protein genes (HvpRPS18 and HvpRPL13) were employed as endogenous controls [40]. Gene-specific primers for 58 neuropeptide precursor genes (Appendix A) were designed using the Primer Premier 5.0 (Premier Biosoft International, Palo Alto, CA, USA), with primer efficiencies (80–120%), standard curves, amplification plots, and melting curves validated prior to expression analysis (Appendix A; Appendix A). Experimental design included triplicate biological replicates. The relative quantification was calculated using the method with two reference genes [41,42,43]. The expression level in the brain sample was used as the calibrator.

### 2.6. Statistical Analysis

Statistical analysis was performed using IBM SPSS21 (IBM Corp., Armonk, NY, USA). Tissue-specific expression data (mean ± SEM) were analyzed by one-way ANOVA with post hoc Tukey’s HSD test. Statistical significance was defined as *p* < 0.05.

## 3. Results

### 3.1. Summary of the Transcriptome Analyses

The RNA-seq dataset yielded 11.78 Gb of raw reads, with 11.55 Gb retained as high-quality clean reads after filtering (Appendix A). The clean reads exhibited Q30 and Q20 scores of 96.62% and 98.95%, respectively (Appendix A). De novo assembly generated 23,040 unigenes with an average length of 1285 bp and a N50 value of 2476 bp (Appendix A). The assembly completeness, assessed via BUSCO analysis, reached 96.1%, comprising 94.7% single-copy orthologs and 1.4% duplicated BUSCOs (Appendix A). Comparing unigenes to six major functional databases for functional annotation, 10,253 (44.50%), 8224 (35.69%), 11,625 (50.46%), 13,599 (59.02%), 8981 (38.98%), and 9534 (41.38%) unigenes had homologous sequences in the GO, KEGG, eggNOG, NR, Swiss-Prot and Pfam, respectively (Appendix A). A Venn diagram revealed 3909 unigenes shared across NR, Swiss-Prot, Pfam, and KEGG databases (Figure 1B). Comparative analysis against the NR database revealed taxonomic distribution patterns of *H. vigintioctopunctata* transcripts, with the highest annotation proportions observed in *Coccinella septempunctata* (43.76%), followed by *Harmonia axyridis* (21.73%), *Tenebrio molitor* (1.46%), and *Tribolium castaneum* (1.18%) (Figure 1C).

Functional annotation classified 10,253 unigenes (44.50%) into 55 Gene Ontology (GO) terms spanning three primary categories: biological processes, cellular components, and molecular functions. Within these categories, the majority of annotated unigenes were associated with the biological processes “metabolic process” and “cellular process”, the cellular components “cell part” and “membrane part”, and the molecular functions “binding” and “catalytic activity” (Appendix A). KEGG pathway analysis assigned 8224 unigenes (35.69%) to six functional groups: Environmental Information Processing, Human Diseases, Organismal Systems, Cellular Processes, Genetic Information Processing, and Metabolism. ‘Signal transduction’ (1083), ‘Cancers: Overview’ (824), ‘Endocrine system’ (565), ‘Transport and catabolism’ (562), ‘Translation’ (456), and ‘Carbohydrate metabolism’ (349) were the dominant pathways in each group, respectively (Figure 2A). In the pathway ‘Nervous system’, 325 unigenes were mapped to 10 KEGG pathways, with “Retrograde endocannabinoid signaling” (112), “Dopaminergic synapse” (88), and “Glutamatergic synapse” (74) representing the most enriched pathways (Figure 2B). In the pathway ‘Endocrine system’, 565 unigenes were annotated to 23 KEGG pathways, dominated by “Thyroid hormone signaling pathway” (98), “Aldosterone synthesis and secretion” (97), and “Oxytocin signaling pathway” (97) (Figure 2C).

### 3.2. Analysis of Neuropeptides

A total of 58 neuropeptide precursor genes were identified in *H. vigintioctopunctata*, classified into 37 evolutionarily conserved families (Table 1). All genes contained complete open reading frames (ORFs), encoding predicted proteins ranging from 69 amino acids (aa) for calcitonin-like diuretic hormone 31 (*CL-DH31*) gene to 371 aa for neuropeptide-like precursor 1 (*NPLP1*) gene, with signal peptide cleavage sites predicted between residues 16 and 34 (Table 1). In silico analysis of post-translational processing predicted 98 unique mature peptides derived from these precursors (Appendix A).

Comparative analysis revealed broad conservation of *H. vigintioctopunctata* neuropeptide precursors across multiple insect species, including *Tribolium castaneum*, *Coccinella septempunctata*, *Tenebrio molitor*, *Grapholitha molesta*, *Harmonia axyridis*, *Carabus violaceus*, and *Zophobas atratus* (Table 1). Most neuropeptide families were successfully identified in *H. vigintioctopunctata* (Appendix A). Among these precursors, part of them were prevalent over species of five different orders (Coleoptera, Diptera, Hemiptera, Hymenoptera, Lepidoptera), such as adipokinetic hormone 2 (AKH 2), allatostatin CC (Ast CC), allatotropin (AT), bursicon, capability (CAPA)/periviscerokinin (PVK)/cardioacceleratory peptide 2b (CAP2b), crustacean cardioactive peptide (CCAP), CCHamide (CCH), Calcitonin-like diuretic hormone 31 (CL-DH31), eclosion hormone (EH), ecdysis triggering hormone (ETH), FMRFamide (FMRF), IDLSRF-like peptide (IDLSRF), insulin-like peptide (ILP), ion transport peptide (ITP), ITG-like (ITG), myosuppressin (MS), Natalisin (NTL), Neuropeptide-like precursor 1 (NPLP1), orcokinin (OK), pigment-dispersing factor (PDF), Pyrokinin (PK)/Pheromone biosynthesis activating neuropeptide (PBAN), ryamide (RY), short neuropeptide F (sNPF), SIFamide (SIF), sulfakinin (SK), and tachykinin (TK) (Appendix A). Notably, AKH/corazonin-related peptide (ACP), allatostatin-C (AstC), elevenin, insulin-like growth factor-like peptide (IGFLP), and IMFamide (IMF) were absent in Coccinellidae species, while corazonin (Crz), allatostatin A (AstA), leucokinin (LK), and LQDVamide (LQDV) remain unreported in Coleoptera genomes (Appendix A). Additionally, calcitonin and corticotropin-releasing factor-like diuretic hormone 47 (CRF-DH47) were not detected in *H. vigintioctopunctata* (Appendix A). We identified six Insulin-like peptides (Table 1). Just one adipokinetic hormone precursor was identified in this species (Table 1). Three groups of osmoregulatory neuropeptide genes were identified in *H. vigintioctopunctata*: One gene encoding an arginine vasopressin-like peptide (AVPL), three genes encoding antidiuretic factors (ADFs), and two genes encoding CRF-like diuretic hormones DH37 and DH44, respectively (Table 1).

Prepropeptide architectures displayed a conserved organization of structural motifs across diverse precursors, such as signal peptides, cleavage sites, and mature peptides. For instance, in AKH, proctolin, SIFamide precursors, and short mature peptides (5–12 aa) were positioned immediately downstream of the signal peptide, followed by longer associated peptides (37–51 aa) (Appendix A). The ecdysis-triggering hormone (ETH) precursor contained two consecutive mature peptides following the signal peptide sequence (Appendix A). Repetitive motifs separated by proteolytic cleavage sites characterized precursors such as FMRFamide (six repeats), allatostatin-B (six), CAPA (two), PBAN (two), and tachykinin (TK) (eight) (Appendix A). Transcript variants were observed in AstCC, CRF-DH37, orcokinin (OK), NPLP1, and neuropeptide F1 (NPF1). Specifically, NPF1 generated two isoforms: a truncated NPF1a lacking a 39-aa fragment present in the full-length NPF1b (Appendix A).

Multi-sequence alignments demonstrated varying identity levels among precursors (Appendix A–S53). Highly-identity sequences included agatoxin (Appendix A), bursicon (Appendix A), ILDSRF (Appendix A), PDF (Appendix A), and Pro (Appendix A), whereas PTTH (Appendix A), PY/PBAN (Appendix A), and tachykinin (TK) (Appendix A) displayed lower sequence identity. Phylogenetic reconstruction confirmed orthology, with all *H. vigintioctopunctata* neuropeptides clustering into clades shared with other insect homologs (Figure 3).

### 3.3. Spatial Expression Patterns of Neuropeptide Precursors

To explore the potential characteristics and roles of these neuropeptides in *H. vigintioctopunctata*, the spatial expression profiles of the 58 neuropeptide precursor genes were examined across larval tissues of the brain, ventral nerve cord (VNC), gut, and Malpighian tubules (MT). Among these, most neuropeptide precursor genes were expressed at higher levels in the brain and VNC tissues. For brain-enriched precursor genes: fourteen genes exhibited predominant expression in the brain, including *AKH*, *EH*, *ILP2*, *ILP3*, *ILP4*, *ILP5*, *MS*, *NTL*, *NPA*, *PDF*, *PTTH*, *SIF*, *SK*, and *TK* (Figure 4A). For VNC-enriched precursor genes: fifteen genes showed elevated expression in the VNC, such as *AstB*, *AstCC-X1*, *AstCC-X2*, *AVPL*, *Burβ*, *CAPA*, *CCAP*, *CCH1*, *CCH2*, *ETH*, *FMRF*, *ILP7*, *NPLP1-X2*, *Pro*, and *PK* (Figure 4B). For brain-VNC co-expressed precursor genes: twenty-two genes were highly expressed in both tissues, including *ALP*, *AstCCC*, *AT*, *Baratin*, *Burα*, *CL-DH31*, *CRF-DH37-X1*, *CRF-DH37-X2*, *GPA2*, *GPB5*, *Hansolin*, *IDLSRF*, *ILP1*, *ITP*, *ITG*, *NPF1b*, *NPLP1-X1*, *OK-A*, *RF*, *RY*, and *sNPF* (Figure 4C). For non-neural tissue expression: *ADF-b1*, *ADF-b4*, and *ADF-b5* were enriched in MT, with additional expression in the brain and VNC (Figure 4D); *CNMa*, *OK-B*, and *TR* showed prominence in gut, while *NPF1a*, *OK-A*, and *sNPF* were also weakly expressed in gut tissue (Figure 4C,D). Multi-tissue expression: *CRF-DH44* was ubiquitously expressed in the brain, VNC, and MT (Figure 4D).

### 3.4. G Protein-Coupled Receptors for Neuropeptides

We identified 31 putative neuropeptide G protein-coupled receptors (GPCRs) in the CNS transcriptome of *H. vigintioctopunctata*, and all have complete ORFs encoding proteins ranging from 249 to 1393 amino acids (aa) (Appendix A). Structural prediction indicated that 30 GPCRs possess ≥ 6 transmembrane domains (TMDs), while the insulin receptor retained a single TMD (Appendix A). Among these GPCRs, 25 receptors belong to Family A, 4 belong to Family B, and 2 belong to leucine-rich repeat-containing GPCRs (LGRs).

Ligand specificity of the 25 Family A receptors was inferred through BLASTP and phylogenetic analyses, resolving them into 19 functional groups (Appendix A): Allatostatin A-like (HvpA1), AKH (HvpA2), Allatotropin (AT) (HvpA3, HvpA4), CAPA/periviscerokinin (PVK) (HvpA5), CCH1 (HvpA6), CCH2 (HvpA7), CCAP (HvpA8, HvpA9), ecdysis triggering hormone (ETH) (HvpA10, HvpA11), FMRF (HvpA12), insulin (HvpA13, HvpA14), MS (HvpA15), NPF (HvpA16), pyrokinin (PK) (HvpA17, HvpA18), RY (HvpA19), sex peptide/allatostatin B (SP/AstB) (HvpA20), TK/ITPL (HvpA21, HvpA22), trissin (TR) (HvpA23), SK (Hvp24), and orphan receptors (HvpA25-NPFF). The phylogenetic analysis reflected clear orthologous relationships within Family A neuropeptide GPCRs (Figure 5A).

For Family B GPCRs, the phylogenetic tree revealed that four receptors were classified into four branches (Figure 5B). HvpB1 clustered with *B. mori* and *Drosophila* DH31 receptors. HvpB2 grouped with *H. axyridis* and *Drosophila* DH44 receptors (Figure 5B). HvpB3 clustered with *B. mori* and *Drosophila* pigment dispersing factor (PDF) receptors. HvpB4 (orphan receptor) exhibited orthology to those of *H. axyridis* and *T. madens* PTH-like receptors (Figure 5B). Here, two LGRs were identified in *H. vigintioctopunctata*, and they were GPA2/GPA5 receptor (HvpLGR1) and bursicon receptor (HvpLGR2) (Appendix A). The phylogenetic tree showed that HvpLGR1 is closely related to the receptor for glycoprotein hormones in *D. melanogaster* (Dme_LGR1) and HvpLGR2 clustered with bursicon receptors from other insects (Figure 5C).

## 4. Discussion

Neuropeptides play critical roles in insect physiology by regulating processes like feeding, reproduction, and behavior through intercellular communication [20,26]. Their essential functions throughout insect life cycles make neuropeptide signaling pathways promising targets for developing selective insecticides with improved environmental profiles. In this study, 58 neuropeptide precursors and 31 neuropeptide receptors were identified by RNA sequencing in the pest 28-spotted ladybird *H. vigintioctopunctata*.

Conventional transcriptome sequencing typically yields 6 Gb per sample. To obtain more complete neuropeptide precursor sequences, we generated 11.78 Gb of raw data. The clean reads Q20 (%) are 98.95% > 90%, and clean reads Q30 (%) are 96.62% > 80%, which indicate the transcriptome results are of good quality. Compared to the NR database, the proportion of *H. vigintioctopunctata* transcripts annotated to different species was higher in *Coccinella septempunctata* and *Harmonia axyridis* than other Coleoptera insects (Figure 1C); this may be because they all belong to the ladybird family of insects.

In total, 58 identified neuropeptide precursors were identified in *H. vigintioctopunctata*, compared to 64 in *T. castaneum* [20,26], 57 in *Grapholita molesta* [18], 50 in *Tenebrio molitor* [21], 17 in *Coccinella septempunctata* [22], and 36 in *Apis mellifera* [44]. Interestingly, Calcitonin, and CRF-DH47 had not been identified in *H. vigintioctopunctata*; this may be because their expression levels are very low at the transcript level. Additionally, ACP, Elevenin, and AstC are absent in Coccinellidae species. ACP is also absent in *Igelater*, *Photinus*, *Aquatica*, and *Leptinotarsa*; Elevenin is also absent in *Pogonus* and *Leptinotarsa*. Crz and AstA were not identified in any Coleoptera databases so far. However, the absence of specific families in Coccinellidae (ACP, Elevenin, AstC) and Coleoptera-wide gaps (Crz and AstA) highlights lineage-specific gene loss or divergence, which may be linked to ecological adaptations.

ILPs play important roles in growth and development, reproduction, stress resistance, and lifespan [10]. ILPs have been found in numerous insect species, and the number of ILPs is significantly different, such as only one in *Locusta migratoria* [45], but the silkworm has 50 insulin-like peptides [28]. Here, in *H. vigintioctopunctata*, six genes encoding ILP were identified (Appendix A). These insulin genes display significant sequence diversity, akin to what has been observed in coleopteran species [26]. ILPs possess some conserved residues (six cysteines, one leucine, and one tyrosine) within the A- and B-chains that are indispensable for tertiary structure formation [46,47] (Appendix A).

The PBAN/Pyrokinin neuropeptide family, with an FXPRLamide motif in the C-terminus, stimulates pheromone biosynthesis, hindgut muscle contraction, diapause, and cuticle melanization [48]. The precursor gene of *PBAN* encodes five neuropeptides including DH, α-SGNP, β-SGNP, γ-SGNP, and PBAN, in moths such as *B. mori* and *Helicoverpa assulta* [49,50]. However, two neuropeptides encoded by the PBAN precursor gene were predicted in *H. vigintioctopunctata* (Appendix A). This possibly indicates the pleiotropic nature of the PBAN/Pyrokin family of peptides across different insects.

The presence of a single AKH precursor (AKH2) in *H. vigintioctopunctata* contrasts with the three AKH paralogs (AKH1, AKH2, AKH3) reported in other insects, such as *Tribolium castaneum*, *Locusta migratoria* and *Grapholita molesta* [18,20,51]. This discrepancy raises intriguing questions about the evolutionary dynamics and functional specialization of *AKH* genes across insect lineages [51,52,53]; they typically exhibit functional diversification through gene duplication in different species. The absence of AKH1 and AKH3 may reflect functional redundancy or reduced selective pressure for specialized paralogs in *H. vigintioctopunctata*’s ecological niche that need further studies to verify.

The expansion of conservation to alternative splicing patterns is a significant finding. The fact that several neuropeptide genes, including *Ast CC*, *CRF-DH 37*, *OK*, *NPLP1*, and *NPF1* have consistent transcript variants in *H. vigintioctopunctata* implies that alternative splicing can generate distinct neuropeptide sequences. Phylogenetic clustering of *H. vigintioctopunctata* neuropeptides with orthologs from diverse insects (e.g., *Tribolium*, *Drosophila*) suggests sequence-level similarities.

The spatial expression profiling of neuropeptide precursor genes in *H. vigintioctopunctata* larvae reveals distinct tissue-specific patterns. The predominant expression of most neuropeptide precursors in the brain and VNC aligns with their canonical functional properties as neurohormones or neuromodulators in central and peripheral nervous systems. For instance, the high expression of *AKH*, *EH*, *PTTH*, and *ILPs* in the brain may correlate with their involvement in critical processes such as energy mobilization [52], ecdysis [54], developmental timing [55], and insulin-like signaling [56]. Similarly, the enrichment of *CAPA*, *CCAP*, and *ETH* in the VNC suggests their potential roles in myotropic effects on heart muscles [57] and ecdysis-related behaviors [58,59].

The co-expression of 22 precursors in both the brain and VNC (e.g., *AT*, *CRF-DH37*, *NPF1b*) implies their dual roles in integrating systemic and local signaling. For example, neuropeptides like *CRF-DH44* (expressed across brain, VNC, and Malpighian tubules) have multiple roles, including regulation of body-fluid secretion, internal nutrient sensing, and CO_2_-dependent response in *Drosophila* [60]. The tissue-restricted expression of *CNMa*, *OK B*, and *TR* in the gut highlights their potential involvement in digestive or peristaltic regulation, while *ADF-b1*/*4*/*5* in Malpighian tubules may contribute to controlling water balance as in *Tenebrio molitor* [61]. Notably, the expression of NPF1a and sNPF in the gut aligns with their known roles in appetite regulation and gut motility in other insects [9]. Overall, the spatial divergence in neuropeptide expression underscores the modularity of peptidergic signaling, enabling precise regulation of tissue-specific processes while maintaining systemic coordination. Future studies could validate these patterns via in situ hybridization or RNAi to clarify functional hierarchies and cross-talk among these neuropeptides in this species.

The identification of 31 putative neuropeptide GPCRs in the central nervous system (CNS) transcriptome of *H. vigintioctopunctata*. Notably, the majority of these receptors belong to Family A (25 out of 31), which aligns with observations in other insects. Phylogenetic analysis and BLASTP-based ligand predictions classified these receptors into 19 distinct groups, including receptors for allatostatins, AKH, CCAP, ETH, and others, underscoring their potential roles in regulating critical processes such as growth, metabolism, ecdysis, and reproduction. Family B GPCRs, though fewer in number, exhibited clear orthology to receptors for diuretic hormones (DH31, DH44) and pigment-dispersing factor (PDF), suggesting they may be involved in osmoregulation, stress responses, and circadian rhythm modulation. These functional inferences are provisional pending ligand-binding and knockdown experiments.

The identification of neuropeptides and their receptor sequences in the pest *H. vigintioctopunctata* provides molecular insights for advancing pest management strategies. Specifically, neuropeptides and their receptors play pivotal regulatory roles in insect growth, development, and behavior, making them promising targets for pest control. For instance, analogs or inhibitors of neuropeptides and their receptors can suppress pest development and increase mortality. CAPA/CAP2b neuropeptides, known to modulate water and ion balance by regulating physiological metabolism, have been shown to induce higher mortality in stink bugs and aphids when applied as analogs [62,63]. Similarly, biostable multi-Aib analogs of tachykinin-related peptides (TRPs), multifunctional neuropeptides widespread in arthropods, exhibit potent oral aphicidal activity in the pea aphid *Acyrthosiphon pisum* [64]. Disruption of insect diapause, a survival strategy to evade adverse seasons, has been achieved using agonists and antagonists of diapause hormone in *Helicoverpa zea* [65]. In recent years, RNA interference (RNAi)-based biopesticides have gained significant attention due to their high specificity and low ecological risk. A notable example is Calantha, a sprayable RNAi formulation developed by GreenLight Biosciences to target the proteasome subunit beta type-5 (*PSMB5*) gene of the Colorado potato beetle (*Leptinotarsa decemlineata*) [66]. RNAi targeting neuropeptides and their receptors has similarly demonstrated potential for pest control. For example, silencing bursicon or its receptor disrupted cuticle tanning and wing expansion, causing lethal developmental defects in *Aphis citricidus* and *Henosepilachna vigintioctomaculata* [67,68]. In *Tribolium castaneum*, RNAi-mediated knockdown of adipokinetic hormones (*AKHs*) and their receptor significantly reduced locomotor activity [69]. Similarly, silencing CCH1 neuropeptide and its receptor in aphids led to reduced feeding and reproductive capacity [70]. These cases underscore the broad applicability of neuropeptide-targeted RNAi in disrupting critical physiological and behavioral processes in pests.

## 5. Conclusions

In general, our study describes a combined strategy of CNS transcriptome of *H. vigintioctopunctata* whose screening resulted in the discovery and identification of its neuropeptide and GPCR genes. These results provide a basis for further pharmacological studies to design mimetic analogs of peptides or antagonists and agonists of receptors for control strategy on this solanaceous crop pest. Future functional studies, including ligand-receptor binding assays, tissue-specific localization, RNA interference, and CRISPR-Cas9 gene editing, will be critical to validate these predictions and unravel the precise roles of these receptors in *H. vigintioctopunctata* physiology. The sequence identification of these neuropeptides and their receptors provides a foundational resource for exploring insect neuroendocrinology and developing targeted strategies for pest management.

## Figures and Tables

**Figure 1 insects-16-00624-f001:**
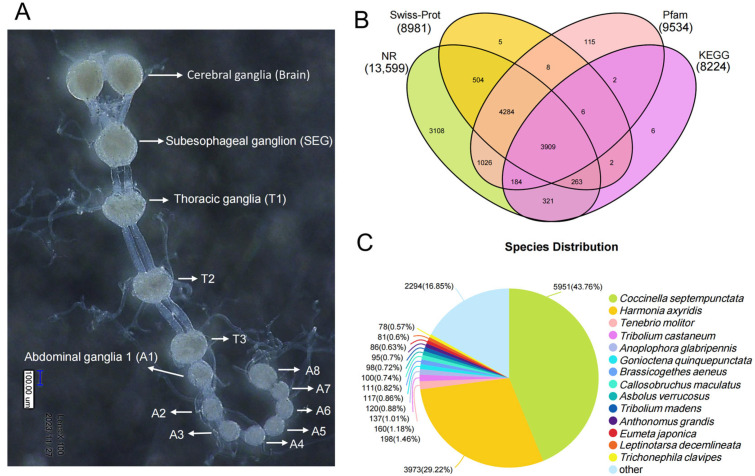
Transcriptomics analysis of *H. vigintioctopunctata* central nervous system (CNS). (**A**) Anatomy of the CNS from *H. vigintioctopunctata.* (**B**) A Venn diagram across NR, Swiss-Prot, Pfam, and KEGG databases. (**C**) Comparative analysis against the NR database revealed taxonomic distribution patterns of *H. vigintioctopunctata* transcripts.

**Figure 2 insects-16-00624-f002:**
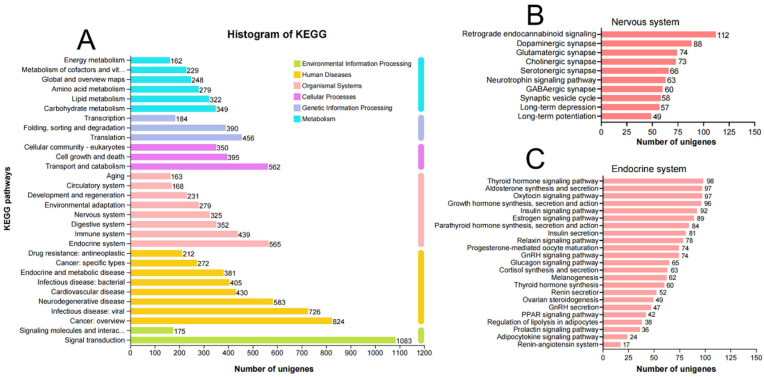
Distribution of transcriptomic unigenes in the KEGG pathways. (**A**) KEGG pathway analysis assigned unigenes to six functional groups: Environmental Information Processing, Human Diseases, Organismal Systems, Cellular Processes, Genetic Information Processing, and Metabolism. (**B**) The KEGG pathways in the pathway ‘Nervous system’. (**C**) The KEGG pathways in the pathway ‘Endocrine system’.

**Figure 3 insects-16-00624-f003:**
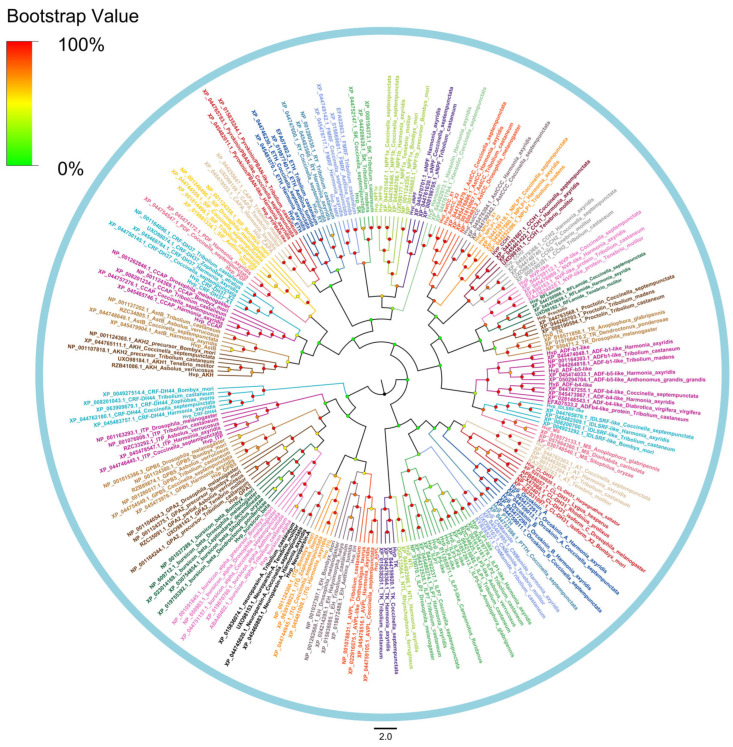
Phylogenetic tree of neuropeptide precursors. The Neighbor-Joining method is used to construct the phylogenetic tree by MEGA11 based on the deduced amino acid sequences of neuropeptide precursors. The different color dots at each node indicate the percentage of bootstrap value after 1000 replications. The GenBank accession numbers is listed in Appendix A.

**Figure 4 insects-16-00624-f004:**
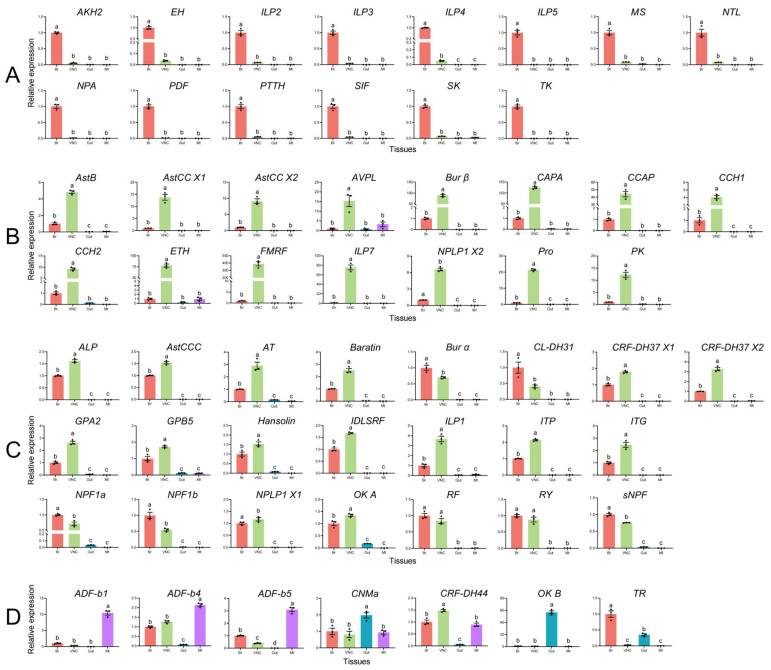
Spatial expression of neuropeptide precursors in *H. vigintioctopunctata.* (**A**) Brain-enriched precursors. (**B**) VNC-enriched precursors. (**C**) Brain–VNC co-expressed precursors. (**D**) Non-neural tissue expression. Br: Brain; VNC: ventral nerve cord (VNC); MT: Malpighian tubes. Data are mean ± standard error (mean ± SE). Different letters indicate significant differences among groups (*p* < 0.05).

**Figure 5 insects-16-00624-f005:**
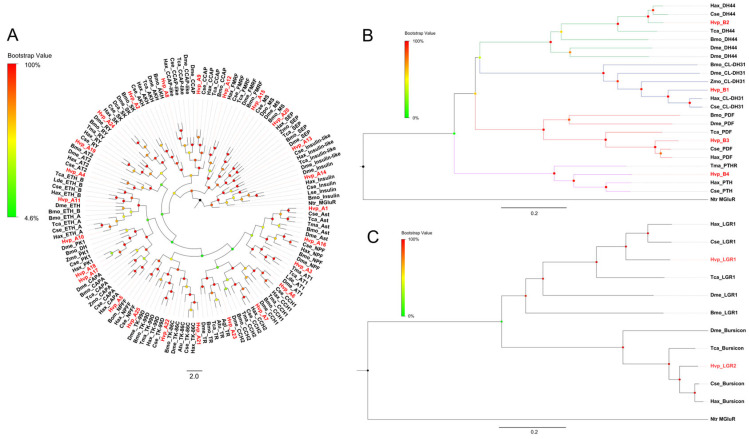
Phylogenetic tree of the Family A neuropeptide G protein-coupled receptors (GPCRs) (**A**), Family B neuropeptide GPCRs (**B**), and leucine-rich repeat-containing GPCRs (**C**). The Neighbor-Joining method is used to construct the phylogenetic tree by MEGA11 based on the deduced amino acid sequences of transmembrane domains 1–7 of GPCRs. The different-colored dots at each node indicate the percentage of bootstrap value after 1000 replications. The GenBank accession numbers are listed in Appendix A.

**Table 1 insects-16-00624-t001:** Neuropeptide precursors identified from *Henosepilachna vigintioctopunctata*.

Classification	Family	Neuropeptide Precursors	GenBank Accession No.	Transcripts Per Million (TPM)	Precursor size (aa)	Signal Peptide (aa)	Homology Search with Known Protein (Blastp)			
							Species	E-Value	Identity (%)	Accession No.
Neuropeptides	Adipokinetic hormone/ Hipertrehalosemic hormone/ Red pigment-concentrating	Adipokinetic hormone 2 (AKH2)	PV645140	116.45	72, complete	20	*Tribolium castaneum*	8.00 × 10^18^	58.90%	NP_001107818.1
Neuropeptides	Agatoxin-like peptide a	Agatoxin-like (ALP)/LQDVamide	PV645141	14.86	106, complete	23	*Carabus violaceus*	1.00 × 10^42^	66.67%	XP_968442.2
Neuropeptides	Allatostatin	Allatostatin B (AstB)/Prothoracicostatic peptide (PTSP)/myoinhibitory peptide (MIP)	PV645142	541.37	192, complete	29	*Tribolium castaneum*	2.00 × 10^50^	48.45%	RZC34805.1
Neuropeptides	Allatostatin	Allatostatin CC (AstCC) X1	PV645143	188.64	137, complete	23	*Zophobas atratus*	1.00 × 10^53^	67.72%	UXO98062.1
Neuropeptides	Allatostatin	Allatostatin CC (AstCC) X2	PV645144	4.2	140, complete	26	*Zophobas atratus*	2.00 × 10^53^	67.72%	UXO98062.1
Neuropeptides	Allatostatin	Allatostatin CCC (AstCCC)	PV645145	215.35	104, complete	23	*Coccinella septempunctata*	9.00 × 10^36^	64%	XP_044759342 [26]
Neuropeptides	Allatotropin/Orexin	Allatotropin (AT)	PV645146	189.20	99, complete	24	*Tribolium castaneum*	1.00 × 10^13^	40.95%	NP_001137204.1
Osmoregulatory neuropeptides	Antidiuretic hormone	Antidiuretic factor b-1 (ADF-b1)	PV645147	14.64	129, complete	22	*Tribolium castaneum*	1.00 × 10^21^	38.00%	EFA07530.1 [20]
Osmoregulatory neuropeptides	Antidiuretic hormone	Antidiuretic factor b-4 (ADF-b4)	PV645148	146.37	159, complete	18	*Tribolium castaneum*	5.00 × 10^33^	57.00%	EFA07533.2 [20]
Osmoregulatory neuropeptides	Antidiuretic hormone	Antidiuretic factor b-5 (ADF-b5)	PV645149	197.04	138, complete	18	*Tribolium castaneum*	2.00 × 10^30^	48.00%	EFA07534.1 [20]
Osmoregulatory neuropeptides	Vasopressin/oxytocin	Arginine-vasopressin-like (AVPL)	PV645150	388.95	152, complete	21	*Tribolium castaneum*	1.00 × 10^55^	61.22%	NP_001078831.1
Other putative neuropeptide genes	NA	Baratin (NVP-like)	PV645151	684.9	314, complete	21	*Tribolium castaneum*	1.00 × 10^80^	51.36%	EFA09163.1 [20]
Protein hormones (polypeptides)	Cystine knot	Bursicon alpha (Bur α)	PP430623	544.05	169, complete	29	*Tribolium castaneum*	4.00 × 10^94^	92.00%	NP_001107779.1
Protein hormones (polypeptides)	Cystine knot	Bursicon beta (Bur β)/partner of bursicon	PP430624	385.62	142, complete	26	*Tribolium castaneum*	3.00 × 10^70^	80.15%	NP_001107780.1
Neuropeptides	Pyrokinin/Periviscerokinin/ Pheromone biosynthesis activating neuropeptide	Capability (CAPA)/Periviscerokinin (PVK)/Cardioacceleratory peptide 2b (CAP2b)	PV645153	598.23	144, complete	19	*Zophobas atratus*	3.00 × 10^09^	37.07%	UXO98070.1
Neuropeptides	Crustacean cardioactive peptide	Crustacean cardioactive peptide (CCAP)	PV645160	423.61	142, complete	19	*Diorhabda sublineata*	3.00 × 10^49^	62.24%	XP_056637896.1
Neuropeptides	CCHamide	CCHamide 1 (CCH1)	PV645154	173.87	161, complete	33	*Tenebrio molitor*	6.00 × 10^23^	37.09%	UXO98161.1
Neuropeptides	CCHamide	CCHamide 2 (CCH2)	PV645155	260.93	118, complete	27	*Tenebrio molitor*	3.00 × 10^27^	41.23%	UXO98162.1
Neuropeptides	CNMamide	CNMamide (CNMa)	PV645156	8.72	142, complete	15	*Tenebrio molitor*	5.00 × 10^12^	34.04%	UXO98163.1
Neuropeptides	Diuretic hormone	Calcitonin-like diuretic hormone 31 (CL-DH31)	PV645152	78.79	69, complete	30	*Tribolium castaneum*	3.00 × 10^07^	61.54%	EEZ99367.2
Osmoregulatory neuropeptides	Corticotropin-releasing hormone binding protein/Diuretic hormone-related	Corticotropin-releasing factor-like diuretic hormone 37 X1 (CRF-DH 37 X1)	PV645157	156.13	136, complete	18	*Tribolium castaneum*	1.00 × 10^12^	34.31%	NP_001164096.1 [26]
Osmoregulatory neuropeptides	Corticotropin-releasing hormone binding protein/Diuretic hormone-related	Corticotropin-releasing factor-like diuretic hormone 37 X2 (CRF-DH 37 X2)	PV645158	92.62	155, complete	18	*Tribolium castaneum*	2.00 × 10^33^	45.57%	XP_015835155.1 [26]
Osmoregulatory neuropeptides	Corticotropin-releasing hormone binding protein/Diuretic hormone-related	Corticotropin-releasing factor-like diuretic hormone 44 (CRF-DH 44)	PV645159	135.5	349, complete	22	*Grapholitha molesta*	1.00 × 10^70^	46.00%	MN639889
Protein hormones (polypeptides)	Eclosion hormone	Eclosion hormone (EH)	PV645162	16.63	81, complete	25	*Halyomorpha halys*	6.00 × 10^28^	66.22%	XP_024214295.1
Protein hormones (polypeptides)	Ecdysis-triggering hormone	Ecdysis-triggering hormone (ETH)	PV645161	26.51	155, complete	22	*Colaphellus bowringi*	1.00 × 10^14^	43.44%	UDO48204.1
Neuropeptides	FMRFamide-related peptide	FMRFamide (FMRF)	PV645163	469.43	207, complete	17	*Harmonia axyridis*	1.00 × 10^62^	52.11%	XP_045478717.1
Protein hormones (polypeptides)	Glycoprotein hormone	Glycoprotein hormone alpha 2 (GPA2)	PV645164	187.93	122, complete	16	*Tribolium castaneum*	4.00 × 10^67^	77.87%	NP_001164244.1
Protein hormones (polypeptides)	Glycoprotein hormone	Glycoprotein hormone beta 5 (GPB5)	PV645165	144.95	154, complete	20	*Tribolium castaneum*	1.00 × 10^65^	64.47%	NP_001280517.1
Neuropeptides	NA	Hansolin	PV645166	5.20	121, complete	20	*Tenebrio molitor*	4.00 × 10^13^	33.61%	UXO98170.1
Neuropeptides	NA	IDLSRF-like peptide (IDLSRF)	PV645167	260.25	208, complete	34	*Coccinella septempunctata*	4.00 × 10^144^	95.19%	XP_044765879.1
Protein hormones (polypeptides)	Insulin/Insulin-like growth factor/Relaxin	Insulin-like peptide 1 (ILP1)	PV645168	16.26	123, complete	21	*Harmonia axyridis*	1.00 × 10^42^	58.68%	XP_045479779.1
Protein hormones (polypeptides)	Insulin/Insulin-like growth factor/Relaxin	Insulin-like peptide 2 (ILP2)	PV645169	122.58	124, complete	21	*Harmonia axyridis*	1.00 × 10^18^	38.64%	XP_045479306.1
Protein hormones (polypeptides)	Insulin/Insulin-like growth factor/Relaxin	Insulin-like peptide 3 (ILP3)	PV645170	13.66	131, complete	20	*Coccinella septempunctata*	1.00 × 10^13^	36.36%	XP_044748903.1
Protein hormones (polypeptides)	Insulin/Insulin-like growth factor/Relaxin	Insulin-like peptide 4 (ILP4)	PV645171	15.62	109, complete	21	*Camponotus floridanus*	5.00 × 10^08^	34.62%	XP_025267327.1
Protein hormones (polypeptides)	Insulin/Insulin-like growth factor/Relaxin	Insulin-like peptide 5 (ILP5)	PV645172	6.54	120, complete	23	*Cryptolaemus montrouzieri*	3.00 × 10^08^	36.59%	KAL3266362.1
Protein hormones (polypeptides)	Insulin/Insulin-like growth factor/Relaxin	Insulin-like peptide 7 (ILP7)/Relaxin	PV645173	79.70	144, complete	24	*Coccinella septempunctata*	8.00 × 10^81^	79.86%	XP_044762237.1
Protein hormones (polypeptides)	Crustacean hyperglycaemic hormone family	ion transport peptide (ITP)	PV645174	48.91	122, complete	28	*Coccinella septempunctata*	3.00 × 10^73^	85.95%	XP_044746445.1
Other putative neuropeptide genes	ITG-like	ITG-like (ITG)	PV645175	850.08	216, complete	20	*Harmonia axyridis*	3.00 × 10^131^	89.30%	XP_045481008.1
Neuropeptides	Myosuppressin	Myosuppressin (MS)	PV645178	2.04	86, complete	24	*Anoplophora glabripennis*	1.00 × 10^26^	57.89%	XP_018573133.1
Neuropeptides	Tachykinin-related peptides	Natalisin (NTL)	PV645179	11.50	150, complete	19	*Rhynchophorus ferrugineus*	4.00 × 10^20^	38.78%	QGA72564.1
Protein hormones (polypeptides)	Neuroparsin/Ovary ecdysteroidogenic hormone	Neuroparsin A (NPA)	PV645180	129.02	101, complete	25	*Harmonia axyridis*	7.00 × 10^32^	62.38%	XP_045460883.1
Neuropeptides	Neuropeptide Y	Neuropeptide F 1a (NPF1a)	PV645211	6.25	88, complete	26	*Tenebrio molitor*	8.00 × 10^21^	57.95%	UXO98177.1
Neuropeptides	Neuropeptide Y	Neuropeptide F 1b (NPF1b)	PV645212	23.98	125, complete	26	*Zophobas atratus*	4.00 × 10^39^	58.73%	UXO98088.1
Other putative neuropeptide genes	Neuropeptide-like precursor	Neuropeptide-like precursor 1 X1 (NPLP1 X1)	PV645209	29.12	371, complete	24	*Coccinella septempunctata*	5.00 × 10^147^	63.71%	XP_044759118.1
Other putative neuropeptide genes	Neuropeptide-like precursor	Neuropeptide-like precursor 1 X2 (NPLP1 X2)	PV645210	4.74	371, complete	24	*Coccinella septempunctata*	2.00 × 10^143^	64.36%	XP_044759118.1
Neuropeptides	Orcokinin	Orcokinin A (OK A)	PV645213	17.87	151, complete	19	*Harmonia axyridis*	7.00 × 10^77^	73.33%	XP_045477291.1
Neuropeptides	Orcokinin	Orcokinin B (OK B)	PV645214	8.59	283, complete	19	*Harmonia axyridis*	4.00 × 10^75^	52.63%	XP_045477290.1
Neuropeptides	Pigment-dispersing hormone/Pigment- dispersing factor	Pigment-dispersing factor (PDF)	PV645215	207.47	105, complete	27	*Photinus pyralis*	3.00 × 10^07^	37.50%	XP_031349268.1
Protein hormones (polypeptides)	NA	Prothoracicotropic hormone (PTTH)	PV645217	15.86	176, complete	18	*Tribolium madens*	2.00 × 10^19^	31.25%	XP_044258306.1
Neuropeptides	Proctolin	Proctolin (Pro)	PV645216	3543.72	86, complete	30	*Rhynchophorus ferrugineus*	4.00 × 10^16^	53.25%	QGA72571.1
Neuropeptides	Pyrokinin/Periviscerokinin/ Pheromone biosynthesis activating neuropeptide	Pyrokinin (PK)/phermone biosynthesis activating neuropeptide like (PBAN-like)	PV645218	467.84	142, complete	23	*Tribolium castaneum*	7.00 × 10^13^	38.32%	XP_015835244.1
Neuropeptides	RFamide neuropeptide	RFLamide (RF)	PV645219	59.54	182, complete	27	*Tenebrio molitor*	1.00 × 10^42^	43.46%	UXO98182.1
Neuropeptides	Luqin/Ryamide	Ryamide (RY)	PV645220	32.54	120, complete	23	*Coccinella septempunctata*	1.00 × 10^38^	63.64%	XP_044747600.1
Neuropeptides	Neuropeptide Y	short neuropeptide F (sNPF)	PV645221	217.92	99, complete	26	*Harmonia axyridis*	5.00 × 10^41^	68.82%	XP_045470701.1
Neuropeptides	FMRFamide related peptide	SIFamide (SIF)	PV645222	130.03	74, complete	25	*Coccinella septempunctata*	1.00 × 10^25^	67.12%	XP_044744559.1
Neuropeptides	Gastrin/cholecystokinin	Sulfakinin (SK)	PV645223	10.41	104, complete	27	*Coccinella septempunctata*	8.00 × 10^25^	49.51%	XP_044752147.1
Neuropeptides	Tachykinin-related peptides	Tachykinin (TK)	PV645224	81.99	276, complete	22	*Coccinella septempunctata*	5.00 × 10^97^	61.25%	XP_044759920.1
Neuropeptides	Trissin	Trissin (TR)	PV645225	10.24	86, complete	21	*Anoplophora glabripennis*	2.00 × 10^14^	50.00%	XP_018571856.1

NA: not applicable.

## Data Availability

The research data are deposited in the article and Appendix A, and the identified sequences have been uploaded to NCBI (GenBank Accession Numbers are listed in Table 1 and Appendix A). Further inquiries can be directed to the corresponding author.

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
