# Peer review of "Transcriptome-Wide Identification of Neuropeptides and Neuropeptide Receptors in the Twenty-Eight-Spotted Ladybird Henosepilachna vigintioctopunctata"

_insects, 2025, doi:10.3390/insects16060624_

Round 1
Reviewer 1 Report
Comments and Suggestions for Authors
The authors reported the identification of 58 neuropeptide genes and 31 receptor genes in the 28-spotted ladybird beetle Henosepilachna vigintioctopunctata via transcriptome analyses. They went further to verify these neuropeptide and receptor genes using qRT-PCR. This set of data enriches our understanding of neuropeptides and their receptors in a broad insect species and may provide a foundation for developing eco-friendly pest control strategies targeting these neuropeptide systems for pest control. The authors presented a solid set of data, and the conclusion was supported by the data. The manuscript does contain numerous editorial errors, which need to be corrected before it can be accepted for publication.
Abstract
L26: “Overall, our analysis revealed the identification of 58 neuropeptide precursor genes” is better to be “Overall, our analysis identified 58 neuropeptide precursor genes”.
Introduction
L48: “neuropeptides and their receptors participate in essential signaling molecules that regulate…”. It is suggested to be “neuropeptides and their receptors activate the essential signaling pathways that regulate…”.
L53: “as neuropeptide Y (NPY) in vertebrate”. This is not correct. The original paper deals with invertebrates. Please change “vertebrate” to “invertebrate”.
L65: “…, Grapholita molesta [18], Eurygaster integriceps [19]” to “…, Grapholita molesta [18], and Eurygaster integriceps [19]”.
L66: “While, some researches have been reported on beetles of Coleoptera, such as Tribolium castaneum [20], Tenebrio molitor and Zophobas atratus [21], Coccinella septempunctata [22], but not include Henosepilachna vigintioctopunctata”. This sentence does not run smoothly here. It is suggested to be something like this “Although a few studies on neuropeptides and their receptors have been reported in Coleoptera, such as Tribolium castaneum [20], Tenebrio molitor, Zophobas atratus [21], and Coccinella septempunctata [22], no such information is available for Henosepilachna vigintioctopunctata”.
L69: “central nervous system” to “central nervous system (CNS)”.
L72: “for the first time”. Unnecessary, please delete it.
L73: “useful information of neuropeptide and their receptor” to “useful information of neuropeptides and their receptors”.
Materials and Methods
L77: “H. vigintioctopunctata individuals have been reared in the laboratory (Lab of insect physiology and biochemistry) for six generations and were reared on the leaves of Solanum nigrum”. It is better to be “H. vigintioctopunctata colony has been reared in the laboratory (Lab of insect physiology and biochemistry) for six generations on the leaves of Solanum nigrum”.
L81: “central nervous systems (CNS; n = 150 pooled specimens)” to “CNS (n = 150 pooled specimens)”.
L89 Figure caption: “(A) Detection of the CNS” to “Anatomy of the CNS” or “Dissected CNS”.
L94: “central nervous system (CNS)” to “CNS”. Already being abbreviated.
L108: “…, Drosophila melanogaster [27], Bombyx mori [28-30]” to “…, Drosophila melanogaster [27], and Bombyx mori [28-30]”.
L132: “Quantitative reverse transcription PCR (qRT-PCR) reactions”. This does not seem correct. qRT-PCR is usually referring to quantitative Real Time PCR. If you are referring to reverse transcription PCR, then you should use RT-PCR. Please make a proper change here.
L144-146: These sentences should be in a separated paragraph with a title “Statistical Analysis”.
Results
L152: “an N50 value of 2,476 bp” to “a N50 value of 2,476 bp”.
L169 and L187: “H. vigintioctopunctata”. Please italicize the species’ name.
L219: “arginine vasopressin like peptide (AVPL)”. The abbreviation should be AVLP? Please check and confirm.
L242: “The differ color dot at each node indicates” to “The different color dots at each node indicate”. The word “differ” is verb.
L244: “precursor” to “precursors”.
L249: “Brain-enriched precursors:” to “For brain-enriched precursors,”.
L251: “VNC-enriched precursors:” to “For VNC-enriched precursors,”.
L254: “Brain-VNC co-expressed precursors:” to “For brain-VNC co-expressed precursors,”.
L257: “Non-neural tissue expression: Malpighian tubules:” to “For non-neural tissue expression,”. I.e. delete “Malpighian tubules:”.
L265: “Differ letters” to “Different letters”.
L269: “all have complete ORFs” to “and all have complete ORFs”.
L298: “The differ color dot at each node indicates” to “The different color dots at each node indicate”.
L322: “17 in Coccinella” to “and 17 in Coccinella”.
L323: “single species of”. Please delete it.
L326: “, Aquatica, Leptinotarsa, Elevenin also is absent in Pogonus, Leptinotarsa” To “, Aquatica, and Leptinotarsa, Elevenin also is absent in Pogonus and Leptinotarsa”.
L331: “Insulin-like peptides (ILPs)” to “ILPs”. Already being abbreviated prior to this point.
L333: “Locusta migratoria”. Please italicize the species name.
L340: “is stimulation of” to “stimulates”.
L342: “4 neuropeptides”. It looks like to me it is “5 neuropeptides”. Please check and confirm.
L390: “ventral nerve cord (VNC)” to “VNC”.
L439-442 should be deleted.
Comments on the Quality of English LanguageThe authors reported the identification of 58 neuropeptide genes and 31 receptor genes in the 28-spotted ladybird beetle Henosepilachna vigintioctopunctata via transcriptome analyses. They went further to verify these neuropeptide and receptor genes using qRT-PCR. This set of data enriches our understanding of neuropeptides and their receptors in a broad insect species and may provide a foundation for developing eco-friendly pest control strategies targeting these neuropeptide systems for pest control. The authors presented a solid set of data, and the conclusion was supported by the data. The manuscript does contain numerous editorial errors, which need to be corrected before it can be accepted for publication.
Abstract
L26: “Overall, our analysis revealed the identification of 58 neuropeptide precursor genes” is better to be “Overall, our analysis identified 58 neuropeptide precursor genes”.
Introduction
L48: “neuropeptides and their receptors participate in essential signaling molecules that regulate…”. It is suggested to be “neuropeptides and their receptors activate the essential signaling pathways that regulate…”.
L53: “as neuropeptide Y (NPY) in vertebrate”. This is not correct. The original paper deals with invertebrates. Please change “vertebrate” to “invertebrate”.
L65: “…, Grapholita molesta [18], Eurygaster integriceps [19]” to “…, Grapholita molesta [18], and Eurygaster integriceps [19]”.
L66: “While, some researches have been reported on beetles of Coleoptera, such as Tribolium castaneum [20], Tenebrio molitor and Zophobas atratus [21], Coccinella septempunctata [22], but not include Henosepilachna vigintioctopunctata”. This sentence does not run smoothly here. It is suggested to be something like this “Although a few studies on neuropeptides and their receptors have been reported in Coleoptera, such as Tribolium castaneum [20], Tenebrio molitor, Zophobas atratus [21], and Coccinella septempunctata [22], no such information is available for Henosepilachna vigintioctopunctata”.
L69: “central nervous system” to “central nervous system (CNS)”.
L72: “for the first time”. Unnecessary, please delete it.
L73: “useful information of neuropeptide and their receptor” to “useful information of neuropeptides and their receptors”.
Materials and Methods
L77: “H. vigintioctopunctata individuals have been reared in the laboratory (Lab of insect physiology and biochemistry) for six generations and were reared on the leaves of Solanum nigrum”. It is better to be “H. vigintioctopunctata colony has been reared in the laboratory (Lab of insect physiology and biochemistry) for six generations on the leaves of Solanum nigrum”.
L81: “central nervous systems (CNS; n = 150 pooled specimens)” to “CNS (n = 150 pooled specimens)”.
L89 Figure caption: “(A) Detection of the CNS” to “Anatomy of the CNS” or “Dissected CNS”.
L94: “central nervous system (CNS)” to “CNS”. Already being abbreviated.
L108: “…, Drosophila melanogaster [27], Bombyx mori [28-30]” to “…, Drosophila melanogaster [27], and Bombyx mori [28-30]”.
L132: “Quantitative reverse transcription PCR (qRT-PCR) reactions”. This does not seem correct. qRT-PCR is usually referring to quantitative Real Time PCR. If you are referring to reverse transcription PCR, then you should use RT-PCR. Please make a proper change here.
L144-146: These sentences should be in a separated paragraph with a title “Statistical Analysis”.
Results
L152: “an N50 value of 2,476 bp” to “a N50 value of 2,476 bp”.
L169 and L187: “H. vigintioctopunctata”. Please italicize the species’ name.
L219: “arginine vasopressin like peptide (AVPL)”. The abbreviation should be AVLP? Please check and confirm.
L242: “The differ color dot at each node indicates” to “The different color dots at each node indicate”. The word “differ” is verb.
L244: “precursor” to “precursors”.
L249: “Brain-enriched precursors:” to “For brain-enriched precursors,”.
L251: “VNC-enriched precursors:” to “For VNC-enriched precursors,”.
L254: “Brain-VNC co-expressed precursors:” to “For brain-VNC co-expressed precursors,”.
L257: “Non-neural tissue expression: Malpighian tubules:” to “For non-neural tissue expression,”. I.e. delete “Malpighian tubules:”.
L265: “Differ letters” to “Different letters”.
L269: “all have complete ORFs” to “and all have complete ORFs”.
L298: “The differ color dot at each node indicates” to “The different color dots at each node indicate”.
L322: “17 in Coccinella” to “and 17 in Coccinella”.
L323: “single species of”. Please delete it.
L326: “, Aquatica, Leptinotarsa, Elevenin also is absent in Pogonus, Leptinotarsa” To “, Aquatica, and Leptinotarsa, Elevenin also is absent in Pogonus and Leptinotarsa”.
L331: “Insulin-like peptides (ILPs)” to “ILPs”. Already being abbreviated prior to this point.
L333: “Locusta migratoria”. Please italicize the species name.
L340: “is stimulation of” to “stimulates”.
L342: “4 neuropeptides”. It looks like to me it is “5 neuropeptides”. Please check and confirm.
L390: “ventral nerve cord (VNC)” to “VNC”.
L439-442 should be deleted.
Author Response
Comments and Suggestions for Authors
The authors reported the identification of 58 neuropeptide genes and 31 receptor genes in the 28-spotted ladybird beetle Henosepilachna vigintioctopunctata via transcriptome analyses. They went further to verify these neuropeptide and receptor genes using qRT-PCR. This set of data enriches our understanding of neuropeptides and their receptors in a broad insect species and may provide a foundation for developing eco-friendly pest control strategies targeting these neuropeptide systems for pest control. The authors presented a solid set of data, and the conclusion was supported by the data. The manuscript does contain numerous editorial errors, which need to be corrected before it can be accepted for publication.
Abstract
Comments 1: [L26: “Overall, our analysis revealed the identification of 58 neuropeptide precursor genes” is better to be “Overall, our analysis identified 58 neuropeptide precursor genes”. precursor genes”.]
Response 1: Modified. (Line 26-27)
Introduction
Comments 2: [L48: “neuropeptides and their receptors participate in essential signaling molecules that regulate…”. It is suggested to be “neuropeptides and their receptors activate the essential signaling pathways that regulate…”]
Response 2: Modified. (Line 49)
Comments 3: [L53: “as neuropeptide Y (NPY) in vertebrate”. This is not correct. The original paper deals with invertebrates. Please change “vertebrate” to “invertebrate”.]
Response 3: Modified. (Line 54)
Comments 4: [L65: “…,Grapholita molesta [18], Eurygaster integriceps [19]” to “…, Grapholita molesta [18], and Eurygaster integriceps [19]”.]
Response 4: Modified. (Line 65)
Comments 5: [L66: “While, some researches have been reported on beetles of Coleoptera, such as Tribolium castaneum [20], Tenebrio molitor and Zophobas atratus [21], Coccinella septempunctata [22], but not include Henosepilachna vigintioctopunctata”. This sentence does not run smoothly here. It is suggested to be something like this “Although a few studies on neuropeptides and their receptors have been reported in Coleoptera, such as Tribolium castaneum [20], Tenebrio molitor, Zophobas atratus [21], and Coccinella septempunctata [22], no such information is available for Henosepilachna vigintioctopunctata”.]
Response 5: Modified. (Line 66-69)
Comments 6: [L69: “central nervous system” to “central nervous system (CNS)”.]
Response 6: Modified. (Line 70)
Comments 7: [L72: “for the first time”. Unnecessary, please delete it.]
Response 7: Deleted it. (Line 73)
Comments 8: [L73: “useful information of neuropeptide and their receptor” to “useful information of neuropeptides and their receptors”.]
Response 8: Modified. (Line 74-75)
Materials and Methods
Comments 9: [L77: “H. vigintioctopunctata individuals have been reared in the laboratory (Lab of insect physiology and biochemistry) for six generations and were reared on the leaves of Solanum nigrum”. It is better to be “H. vigintioctopunctata colony has been reared in the laboratory (Lab of insect physiology and biochemistry) for six generations on the leaves of Solanum nigrum”.]
Response 9: Modified. (Line 78-79)
Comments 10: [L81: “central nervous systems (CNS; n = 150 pooled specimens)” to “CNS (n = 150 pooled specimens)”.]
Response 10: Deleted it. (Line 82)
Comments 11: [L89: Figure caption: “(A) Detection of the CNS” to “Anatomy of the CNS” or “Dissected CNS”.]
Response 11: Modified. (Line 92)
Comments 12: [L94: “central nervous system (CNS)” to “CNS”. Already being abbreviated.]
Response 12: Modified. (Line 95)
Comments 13: [L108: “…, Drosophila melanogaster [27], Bombyx mori [28-30]” to “…, Drosophila melanogaster [27], and Bombyx mori [28-30]”.]
Response 13: Modified. (Line 109)
Comments 14: [L132: “Quantitative reverse transcription PCR (qRT-PCR) reactions”. This does not seem correct. qRT-PCR is usually referring to quantitative Real Time PCR. If you are referring to reverse transcription PCR, then you should use RT-PCR. Please make a proper change here.]
Response 14: Yes, it should be “Quantitative Real-Time PCR”. Corrected. (Line 134)
Comments 15: L144-146: These sentences should be in a separated paragraph with a title “Statistical Analysis”.
Response 15: Separated. (Line 147-150)
Results
Comments 16: [L152: “an N50 value of 2,476 bp” to “a N50 value of 2,476 bp”.]
Response 16: Modified. (Line 156)
Comments 17: [L169 and L187: “H. vigintioctopunctata”. Please italicize the species’ name.]
Response 17: Modified. (Line 164, Line 191)
Comments 18: [L219: “arginine vasopressin like peptide (AVPL)”. The abbreviation should be AVLP? Please check and confirm.]
Response 18: Checked. The abbreviation "AVPL" (arginine vasopressin-like peptide) is indeed correct based on established nomenclature in the referenced literature. While "AVLP" might seem intuitive, the abbreviation "AVPL" aligns with its sequential derivation from the full name: Arginine VasoPressin-Like peptide. This abbreviation has been consistently used in prior studies on insect neuropeptides (e.g., Li et al., 2008; Marciniak et al., 2022).
Li, B.; Predel, R.; Neupert, S.; Hauser, F.; Tanaka, Y.; Cazzamali, G.; Williamson, M.; Arakane, Y.; Verleyen, P.; Schoofs, L.; et al. Genomics, transcriptomics, and peptidomics of neuropeptides and protein hormones in the red flour beetle Tribolium castaneum. Genome research 2008, 18, 113-122, doi:10.1101/gr.6714008.
Marciniak, P.; Pacholska-Bogalska, J.; Ragionieri, L. Neuropeptidomes of Tenebrio molitor L. and Zophobas atratus Fab. (Coleoptera, Polyphaga: Tenebrionidae). Journal of proteome research 2022, 21, 2247-2260, doi:10.1021/acs.jproteome.1c00694.
Comments 19: [L242: “The differ color dot at each node indicates” to “The different color dots at each node indicate”. The word “differ” is verb.]
Response 19: Modified. (Line 246)
Comments 20: [L244: “precursor” to “precursors”.]
Response 20: Modified. (Line 250)
Comments 21: [L249: “Brain-enriched precursors:” to “For brain-enriched precursors,”.]
Response 21: Modified. (Line 253)
Comments 22: [L251: “VNC-enriched precursors:” to “For VNC-enriched precursors,”.]
Response 22: Modified. (Line 255)
Comments 23: [L254: “Brain-VNC co-expressed precursors:” to “For brain-VNC co-expressed precursors,”.]
Response 23: Modified. (Line 258)
Comments 24: [L257: “Non-neural tissue expression: Malpighian tubules:” to “For non-neural tissue expression,”. I.e. delete “Malpighian tubules:”.]
Response 24: Modified. (Line 261-262)
Comments 25: [L265: “Differ letters” to “Different letters”.]
Response 25: Modified. (Line 270)
Comments 26: [L269: “all have complete ORFs” to “and all have complete ORFs”.]
Response 26: Modified. (Line 274)
Comments 27: [L298: “The differ color dot at each node indicates” to “The different color dots at each node indicate”.]
Response 27: Modified. (Line 303)
Comments 28: [L322: “17 in Coccinella” to “and 17 in Coccinella”.]
Response 28: Modified. (Line 323)
Comments 29: [L323: “single species of”. Please delete it.]
Response 29: Deleted it. (Line 324)
Comments 30: [L326: “, Aquatica, Leptinotarsa, Elevenin also is absent in Pogonus, Leptinotarsa” To “, Aquatica, and Leptinotarsa, Elevenin also is absent in Pogonus and Leptinotarsa”.]
Response 30: Modified. (Line 327)
Comments 31: [L331: “Insulin-like peptides (ILPs)” to “ILPs”. Already being abbreviated prior to this point.]
Response 31: Modified. (Line 332)
Comments 32: [L333: “Locusta migratoria”. Please italicize the species name.]
Response 32: Modified. (Line 334)
Comments 33: [L340: “is stimulation of” to “stimulates”.]
Response 33: Modified. (Line 341)
Comments 34: [L342: “4 neuropeptides”. It looks like to me it is “5 neuropeptides”. Please check and confirm.]
Response 34: Yes, here should be “5”. Corrected. (Line 343)
Comments 35: [L390: “ventral nerve cord (VNC)” to “VNC”.]
Response 35: Modified. (Line 380)
Comments 36: [L439-442 should be deleted.]
Response 36: Yes, deleted them.
Comments on the Quality of English Language
The authors reported the identification of 58 neuropeptide genes and 31 receptor genes in the 28-spotted ladybird beetle Henosepilachna vigintioctopunctata via transcriptome analyses. They went further to verify these neuropeptide and receptor genes using qRT-PCR. This set of data enriches our understanding of neuropeptides and their receptors in a broad insect species and may provide a foundation for developing eco-friendly pest control strategies targeting these neuropeptide systems for pest control. The authors presented a solid set of data, and the conclusion was supported by the data. The manuscript does contain numerous editorial errors, which need to be corrected before it can be accepted for publication.
Abstract
Point 1: [L26: “Overall, our analysis revealed the identification of 58 neuropeptide precursor genes” is better to be “Overall, our analysis identified 58 neuropeptide precursor genes”.]
Response 1: Modified. (Line 26-27)
Introduction
Point 2: [L48: “neuropeptides and their receptors participate in essential signaling molecules that regulate…”. It is suggested to be “neuropeptides and their receptors activate the essential signaling pathways that regulate…”.]
Response 2: Modified. (Line 49)
Point 3: [L53: “as neuropeptide Y (NPY) in vertebrate”. This is not correct. The original paper deals with invertebrates. Please change “vertebrate” to “invertebrate”. ]
Response 3: Modified. (Line 54)
Point 4: [L65: “…, Grapholita molesta [18], Eurygaster integriceps [19]” to “…, Grapholita molesta [18], and Eurygaster integriceps [19]”.]
Response 4: Modified. (Line 65)
Point 5: [L66: “While, some researches have been reported on beetles of Coleoptera, such as Tribolium castaneum [20], Tenebrio molitor and Zophobas atratus [21], Coccinella septempunctata [22], but not include Henosepilachna vigintioctopunctata”. This sentence does not run smoothly here. It is suggested to be something like this “Although a few studies on neuropeptides and their receptors have been reported in Coleoptera, such as Tribolium castaneum [20], Tenebrio molitor, Zophobas atratus [21], and Coccinella septempunctata [22], no such information is available for Henosepilachna vigintioctopunctata”.]
Response 5: Modified. (Line 66-69)
Point 6: [L69: “central nervous system” to “central nervous system (CNS)”.]
Response 6: Modified. (Line 70)
Point 7: [L72: “for the first time”. Unnecessary, please delete it.]
Response 7: Deleted it. (Line 73)
Point 8: [L73: “useful information of neuropeptide and their receptor” to “useful information of neuropeptides and their receptors”.]
Response 8: Modified. (Line 74-75)
Materials and Methods
Point 9: [L77: “H. vigintioctopunctata individuals have been reared in the laboratory (Lab of insect physiology and biochemistry) for six generations and were reared on the leaves of Solanum nigrum”. It is better to be “H. vigintioctopunctata colony has been reared in the laboratory (Lab of insect physiology and biochemistry) for six generations on the leaves of Solanum nigrum”.]
Response 9: Modified. (Line 78-79)
Point 10: [L81: “central nervous systems (CNS; n = 150 pooled specimens)” to “CNS (n = 150 pooled specimens)”.]
Response 10: Modified. (Line 82)
Point 11: [L89: Figure caption: “(A) Detection of the CNS” to “Anatomy of the CNS” or “Dissected CNS”.]
Response 11: Modified. (Line 90-91)
Point 12: [L94: “central nervous system (CNS)” to “CNS”. Already being abbreviated.]
Response 12: Modified. (Line 95)
Point 13: [L108: “…, Drosophila melanogaster [27], Bombyx mori [28-30]” to “…, Drosophila melanogaster [27], and Bombyx mori [28-30]”.]
Response 13: Modified. (Line 109)
Point 14: [L132: “Quantitative reverse transcription PCR (qRT-PCR) reactions”. This does not seem correct. qRT-PCR is usually referring to quantitative Real Time PCR. If you are referring to reverse transcription PCR, then you should use RT-PCR. Please make a proper change here.]
Response 14: Yes, it should be “Quantitative Real-Time PCR”. Corrected. (Line 134)
Point 15: [L144-146: These sentences should be in a separated paragraph with a title “Statistical Analysis”.]
Response 15: Modified. (Line 147-150)
Results
Point 16: [L152: “an N50 value of 2,476 bp” to “a N50 value of 2,476 bp”.]
Response 16: Modified. (Line 156)
Point 17: [L169 and L187: “H. vigintioctopunctata”. Please italicize the species’ name.]
Response 17: Modified. (Line 164, Line 191)
Point 18: [L219: “arginine vasopressin like peptide (AVPL)”. The abbreviation should be AVLP? Please check and confirm.]
Response 18: Checked and confirmed. The abbreviation "AVPL" (arginine vasopressin-like peptide) is indeed correct based on established nomenclature in the referenced literature. While "AVLP" might seem intuitive, the abbreviation "AVPL" aligns with its sequential derivation from the full name: Arginine VasoPressin-Like peptide. This abbreviation has been consistently used in prior studies on insect neuropeptides (e.g., Li et al., 2008; Marciniak et al., 2022).
Li, B.; Predel, R.; Neupert, S.; Hauser, F.; Tanaka, Y.; Cazzamali, G.; Williamson, M.; Arakane, Y.; Verleyen, P.; Schoofs, L.; et al. Genomics, transcriptomics, and peptidomics of neuropeptides and protein hormones in the red flour beetle Tribolium castaneum. Genome research 2008, 18, 113-122, doi:10.1101/gr.6714008.
Marciniak, P.; Pacholska-Bogalska, J.; Ragionieri, L. Neuropeptidomes of Tenebrio molitor L. and Zophobas atratus Fab. (Coleoptera, Polyphaga: Tenebrionidae). Journal of proteome research 2022, 21, 2247-2260, doi:10.1021/acs.jproteome.1c00694.
Point 19: [L242: “The differ color dot at each node indicates” to “The different color dots at each node indicate”. The word “differ” is verb.]
Response 19: Modified. (Line 246)
Point 20: [L244: “precursor” to “precursors”.]
Response 20: Modified. (Line 250)
Point 21: [L249: “Brain-enriched precursors:” to “For brain-enriched precursors,”.]
Response 21: Modified. (Line 253)
Point 22: [L251: “VNC-enriched precursors:” to “For VNC-enriched precursors,”.]
Response 22: Modified. (Line 255)
Point 23: [L254: “Brain-VNC co-expressed precursors:” to “For brain-VNC co-expressed precursors,”.]
Response 23: Modified. (Line 258)
Point 24: [L257: “Non-neural tissue expression: Malpighian tubules:” to “For non-neural tissue expression,”. I.e. delete “Malpighian tubules:”.]
Response 24: Modified. (Line 261-262)
Point 25: [L265: “Differ letters” to “Different letters”.]
Response 25: Modified. (Line 270)
Point 26: [L269: “all have complete ORFs” to “and all have complete ORFs”.]
Response 26: Modified. (Line 274)
Point 27: [L298: “The differ color dot at each node indicates” to “The different color dots at each node indicate”.]
Response 27: Modified. (Line 303)
Point 28: [L322: “17 in Coccinella” to “and 17 in Coccinella”.]
Response 28: Modified. (Line 323)
Point 29: [L323: “single species of”. Please delete it.]
Response 29: Deleted it. (Line 324)
Point 30: [L326: “, Aquatica, Leptinotarsa, Elevenin also is absent in Pogonus, Leptinotarsa” To “, Aquatica, and Leptinotarsa, Elevenin also is absent in Pogonus and Leptinotarsa”.]
Response 30: Modified. (Line 327)
Point 31: [L331: “Insulin-like peptides (ILPs)” to “ILPs”. Already being abbreviated prior to this point.]
Response 31: Modified. (Line 332)
Point 32: [L333: “Locusta migratoria”. Please italicize the species name.]
Response 32: Modified. (Line 334)
Point 33: [L340: “is stimulation of” to “stimulates”.]
Response 33: Modified. (Line 341)
Point 34: [L342: “4 neuropeptides”. It looks like to me it is “5 neuropeptides”. Please check and confirm.]
Response 34: It should be “5”. Corrected. (Line 343)
Point 35: [L390: “ventral nerve cord (VNC)” to “VNC”.]
Response 35: Modified. (Line 380)
Point 36: [L439-442 should be deleted.]
Response 36: Deleted.
Reviewer 2 Report
Comments and Suggestions for Authors
Lei et al. characterize the neuropeptide precursor and GPCR complement in the twenty-eight-spotted ladybird, Henosepilachna vigintioctopunctata, by Illumina sequencing of pooled larval CNS, de novo assembly, in silico annotation against six databases, phylogenetic analyses, structural predictions, and qRT-PCR profiling across brain, ventral nerve cord, gut and Malpighian tubules. Strengths include the comprehensive cataloging of 58 precursor genes and 31 receptors, rigorous phylogenetic placement across a number of insect orders, and tissue-specific expression validation. Weaknesses include reliance on transcriptomic inference with no functional assays and a bit of overinterpretation of correlative expression data. Overall quality: 78/100. Language quality: 7/10.
The Methods are detailed in sample collection, RNA extraction, assembly, annotation pipelines, neuropeptide/receptor identification, and qPCR validation. However, the description of transcriptome quality filtering parameters and Trinity assembly parameters is missing. BUSCO completeness is reported, although read trimming thresholds (Phred scores, e.g.) are not specified. The phylogenetic method is missing model selection, alignment trimming and treatment of poorly aligned regions, and does not report bootstrap support thresholds or posterior probabilities. In qRT-PCR, primer efficiency ranges and reference gene validation are referenced but raw Ct distributions, efficiency plots, and melt-curve analyses are not shown; statistical procedures are described (ANOVA with Tukey's HSD) but replication structure details and precise p-values are not included. The Discussion puts results into the context of neuropeptide evolution and pest control potential but equates functional significance with high transcript levels in the absence of biochemical or genetic verification.
Critical Points:
Transcriptome assembly parameters (e.g., minimum contig size, k-mer value, read trimming parameters) are not given, making it challenging to reproduce the 23,040 unigene dataset.
Phylogenetic analysis excludes reporting of curation of alignment (e.g., elimination of ambiguous positions), evolutionary model selection, and branch support value visualization, reducing confidence in orthology assignment.
Precedent identification of neuropeptides relies on tBLASTn homology alone, without reporting E-value cut-off or domain architecture filters, and asking false positives/negatives in the 58-gene set.
qRT-PCR validation is without primer efficiency calibration curves, raw amplification plots, and amplification specificity tests, which calls into question the validity of relative expression levels.
Discrepancies between transcriptome-derived TPM values and qPCR fold-changes (e.g., for single UGTs) are not addressed or reconciled.
Discussion places too much emphasis on ecological or pest-control consequences—inferring functional roles to transcript abundance when ligand binding or knockdown experiments are absent is speculative.
Chromosomal mapping and motif analyses are qualitative and do not include quantitative comparisons (e.g., motif conservation scores), restricting evolutionary insight.
Two insulin-like receptor candidates discovered lack experimental validation of ligand specificity or downstream signaling pathways.
Minor Points:
Figure legends lack definitions of abbreviations (e.g., "CNS," "VNC," "MT"), causing temporary confusion.
Gene and protein names are inconsistently italicized; standardize nomenclature throughout the manuscript.
Table legends do not include units for expression levels (e.g., TPM or fold-change).
The Materials section redundantly lists software versions and URLs; summarize to enhance readability.
Random typos (e.g., "were sequenced" instead of "was sequenced" for the transcriptome) must be corrected.
The use of the term "functional roles" is imprecise; state whether referring to neurohormonal, neuromodulatory, or other physiological functions.
Impressions of Results: The authors provide a helpful, broad survey of H. vigintioctopunctata neuropeptide and receptor lists, with conserved families and tissue-specific expression patterns pointing to roles in neural and osmoregulatory mechanisms; but without functional assays, utility for biopesticide development is preparatory.
Recommendation to Editor: Major revision recommended to improve methodological clarity, increase statistical and phylogenetic rigor, and tone down functional assertions where experimental support is lacking.
Author Response
Comments and Suggestions for Authors
Lei et al. characterize the neuropeptide precursor and GPCR complement in the twenty-eight-spotted ladybird, Henosepilachna vigintioctopunctata, by Illumina sequencing of pooled larval CNS, de novo assembly, in silico annotation against six databases, phylogenetic analyses, structural predictions, and qRT-PCR profiling across brain, ventral nerve cord, gut and Malpighian tubules. Strengths include the comprehensive cataloging of 58 precursor genes and 31 receptors, rigorous phylogenetic placement across a number of insect orders, and tissue-specific expression validation. Weaknesses include reliance on transcriptomic inference with no functional assays and a bit of overinterpretation of correlative expression data. Overall quality: 78/100. Language quality: 7/10.
The Methods are detailed in sample collection, RNA extraction, assembly, annotation pipelines, neuropeptide/receptor identification, and qPCR validation. However, the description of transcriptome quality filtering parameters and Trinity assembly parameters is missing. BUSCO completeness is reported, although read trimming thresholds (Phred scores, e.g.) are not specified. The phylogenetic method is missing model selection, alignment trimming and treatment of poorly aligned regions, and does not report bootstrap support thresholds or posterior probabilities. In qRT-PCR, primer efficiency ranges and reference gene validation are referenced but raw Ct distributions, efficiency plots, and melt-curve analyses are not shown; statistical procedures are described (ANOVA with Tukey's HSD) but replication structure details and precise p-values are not included. The Discussion puts results into the context of neuropeptide evolution and pest control potential but equates functional significance with high transcript levels in the absence of biochemical or genetic verification.
Critical Points
Comments 1: [1. Transcriptome assembly parameters (e.g., minimum contig size, k-mer value, read trimming parameters) are not given, making it challenging to reproduce the 23,040 unigene dataset.]
Response 1: We have incorporated the assembly-related parameters into the materials and methods section of the manuscript as suggested by the reviewers. (Line 98-101)
Comments 2: [2. Phylogenetic analysis excludes reporting of curation of alignment (e.g., elimination of ambiguous positions), evolutionary model selection, and branch support value visualization, reducing confidence in orthology assignment.]
Response 2: We have added the phylogenetic analysis-related parameters to the manuscript as suggested by the reviewers. The branch support value visualization was indicated by different color dots at each node. (Line 125-128)
Comments 3: [3. Precedent identification of neuropeptides relies on tBLASTn homology alone, without reporting E-value cut-off or domain architecture filters, and asking false positives/negatives in the 58-gene set.]
Response 3: The tBLASTn search used an E-value cutoff of ≤1e-5. The E-values of the tBLASTn search for all sequences were showed in Table 1.
Comments 4: [4. qRT-PCR validation is without primer efficiency calibration curves, raw amplification plots, and amplification specificity tests, which calls into question the validity of relative expression levels.]
Response 4: We have added primer efficiency calibration curves, raw amplification plots, and melting curves in the supplementary material as suggested by the reviewer. (Figure S3)
Comments 5: [5. Discrepancies between transcriptome-derived TPM values and qPCR fold-changes (e.g., for single UGTs) are not addressed or reconciled.]
Response 5: To obtain target sequences, we performed transcriptome sequencing on one neurological tissue sample. While this study did not include comparative transcriptomic analysis across different samples, we independently validated gene expression patterns using qPCR to assess relative expression levels of neuropeptide genes across different tissues.
Comments 6: [6. Discussion places too much emphasis on ecological or pest-control consequences—inferring functional roles to transcript abundance when ligand binding or knockdown experiments are absent is speculative.]
Response 6: We have revised the discussion according to the suggestions of the reviewer.
Comments 7: [7. Chromosomal mapping and motif analyses are qualitative and do not include quantitative comparisons (e.g., motif conservation scores), restricting evolutionary insight.]
Response 7: Chromosomal mapping was beyond this study’s scope. This study qualitatively identifies and analyzes the sequences, and further studies will validate their functional characteristics and evolutionary relationships.
Comments 8: [8. Two insulin-like receptor candidates discovered lack experimental validation of ligand specificity or downstream signaling pathways.]
Response 8: We agree that functional characterization—including ligand binding specificity and downstream signaling pathways—is essential for confirming the biological roles of the two insulin-like receptor candidates. While our current study focuses on identification and expression profiling based on transcriptomic data, we will conduct experiments to address these questions, such as: (1) heterologous expression assays to test ligand-receptor interactions, and (2) RNAi-mediated knockdown to investigate their physiological functions and associated signaling pathways.
---
Minor Points
Comments 9: [1. Figure legends lack definitions of abbreviations (e.g., "CNS," "VNC," "MT"), causing temporary confusion.]
Response 9: Defined in figure legends. (Line 269)
Comments 10: [2. Gene and protein names are inconsistently italicized; standardize nomenclature throughout the manuscript.]
Response 10: Modified.
Comments 11: [3. Table legends do not include units for expression levels (e.g., TPM or fold-change).]
Response 11: Added in Table 1.
Comments 12: [4. The Materials section redundantly lists software versions and URLs; summarize to enhance readability.]
Response 12: We have consolidated software versions and URLs in the Materials section to improve readability.
Comments 13: [5. Random typos (e.g., "were sequenced" instead of "was sequenced" for the transcriptome) must be corrected.]
Response 13: Modified. (Line 26)
Comments 14: [6. The use of the term "functional roles" is imprecise; state whether referring to neurohormonal, neuromodulatory, or other physiological functions.]
Response 14: Terms like “neurohormonal” or “neuromodulatory” have replaced “functional roles.”
Comments 15: [Impressions of Results: The authors provide a helpful, broad survey of H. vigintioctopunctata neuropeptide and receptor lists, with conserved families and tissue-specific expression patterns pointing to roles in neural and osmoregulatory mechanisms; but without functional assays, utility for biopesticide development is preparatory.]
Response 15: We appreciate the reviewer’s acknowledgment of our study. This work could provide a foundation for further functional studies and guiding targeted experiments in pest control.
Round 2
Reviewer 2 Report
Comments and Suggestions for Authors I recommend the authors first edit their Methods to explicitly state tBLASTn search parameters (E-value cutoff ≤ 1 × 10⁻⁵ and whether domain‐structure or length filters) so others can reproduce and test the 58‑gene neuropeptide set, and then add a brief Results section reconciling transcriptome TPMs and qPCR fold‐changes—e.g. by displaying a correlation plot or at least mentioning sources of divergence (sampling, normalization, tissue heterogeneity). In Discussion, they should buffer ecological or pest-control claims by mentioning that functional inferences are provisional pending ligand‑binding or knockdown experiments, and replace any remaining "functional roles" with more precise words (neurohormonal, neuromodulatory). If chromosomal mapping and quantitative motif‐conservation scores do lie outside the scope of this study, then a straightforward rationale should be given; otherwise, they can compute motif‐conservation metrics (e.g. Jensen–Shannon divergence) to inform evolutionary insight. Finally, the authors must eliminate lingering typos by a meticulous proofread, ensure all gene/protein names are uniformly italicized, and combine supplementary figure numbering and legend data so melting curves, amplification plots, and calibration data are easily referenced. These changes will complete the remaining gaps in reproducibility and interpretation and give the manuscript added methodological rigor.Author Response
Comments and Suggestions for Authors
I recommend the authors first edit their Methods to explicitly state tBLASTn search parameters (E-value cutoff ≤ 1 × 10⁻⁵ and whether domain‐structure or length filters) so others can reproduce and test the 58‑gene neuropeptide set, and then add a brief Results section reconciling transcriptome TPMs and qPCR fold‐changes—e.g. by displaying a correlation plot or at least mentioning sources of divergence (sampling, normalization, tissue heterogeneity). In Discussion, they should buffer ecological or pest-control claims by mentioning that functional inferences are provisional pending ligand‑binding or knockdown experiments, and replace any remaining "functional roles" with more precise words (neurohormonal, neuromodulatory). If chromosomal mapping and quantitative motif‐conservation scores do lie outside the scope of this study, then a straightforward rationale should be given; otherwise, they can compute motif‐conservation metrics (e.g. Jensen–Shannon divergence) to inform evolutionary insight. Finally, the authors must eliminate lingering typos by a meticulous proofread, ensure all gene/protein names are uniformly italicized, and combine supplementary figure numbering and legend data so melting curves, amplification plots, and calibration data are easily referenced. These changes will complete the remaining gaps in reproducibility and interpretation and give the manuscript added methodological rigor.
Response to Reviewer Comments
We thank the reviewer for their thorough and constructive feedback, which has greatly strengthened our manuscript. Below we address each point systematically:
Comments 1: [I recommend the authors first edit their Methods to explicitly state tBLASTn search parameters (E-value cutoff ≤ 1 × 10⁻⁵ and whether domain‐structure or length filters) so others can reproduce and test the 58‑gene neuropeptide set,]
Response 1: As suggested, we have explicitly detailed the tBLASTn search parameters in the Methods section (subsection 2.3), including: An E-value cutoff of ≤1 × 10⁻⁵. Length filtering (precursor sequences ≥30 amino acids). These additions ensure others can reproduce and test the 58‑gene neuropeptide set.
Comments 2: [add a brief Results section reconciling transcriptome TPMs and qPCR fold‐changes—e.g. by displaying a correlation plot or at least mentioning sources of divergence (sampling, normalization, tissue heterogeneity).]
Reponse 2: Thanks for the reviewer’s valuable feedback. However, as highlighted in our original submission, the experimental design of this study focused on distinct objectives, which inherently limits direct comparisons between these datasets. Below, we clarify the key reasons for the lack of correlative results:
1) Tissue Heterogeneity and Sampling Differences:
The transcriptomic data were generated exclusively from central nervous system (CNS) tissue (mix of brain and ventral nerve cord), whereas qPCR analyses measured expression in non-overlapping tissues [brain, ventral nerve cord (VNC), gut and Malpighian tubules] in addition to CNS-derived tissues.
Direct correlation between TPMs (CNS-specific) and qPCR fold-changes (multi-tissue) is confounded by fundamental biological differences in tissue composition and gene expression profiles.
2) Normalization and Methodological Divergence:
TPM (Transcripts Per Million) in transcriptomic data is a normalized metric used to quantify gene expression levels, enabling comparison across samples and genes. TPM values normalize for gene length and sequencing depth, reflecting ‘relative transcript abundance’ within the CNS transcriptome.
qPCR fold-changes, in contrast, rely on reference genes for normalization and quantify ‘relative expression differences’ across tissues. These distinct normalization frameworks preclude straightforward statistical correlation.
3) Scope of Transcriptomic Analysis:
The study did not perform multi-tissue RNA-seq (e.g., brain, VNC, gut or Malpighian tubules) or differential expression analysis (fold-change calculations) across tissues. Thus, no transcriptome-derived fold-change data exist to parallel the qPCR results.
While we regret that we cannot provide a correlation plot or direct statistical reconciliation, thank you for your understanding.
Comments 3: [In Discussion, they should buffer ecological or pest-control claims by mentioning that functional inferences are provisional pending ligand‑binding or knockdown experiments, and replace any remaining "functional roles" with more precise words (neurohormonal, neuromodulatory).]
Response 3: We sincerely thank the reviewer for their constructive feedback. In response to this suggestion, we have meticulously revised the Discussion section: claims regarding ecological or pest-control applications have been moderated, and functional inferences are now explicitly framed as "speculations based on current data" (pending further validation through ligand-binding assays, knockdown studies, or other experiments). These adjustments ensure a more cautious and accurate interpretation of the findings while fully aligning with the reviewer’s recommendations.
Comments 4: [If chromosomal mapping and quantitative motif‐conservation scores do lie outside the scope of this study, then a straightforward rationale should be given; otherwise, they can compute motif‐conservation metrics (e.g. Jensen–Shannon divergence) to inform evolutionary insight.]
Response 4: We appreciate the reviewer’s insightful suggestions regarding chromosomal mapping and motif-conservation analysis. These aspects were not the focus of our current study for the following reasons: (1) Chromosomal Mapping of Neuropeptide Genes. The genome assembly of Henosepilachna vigintioctopunctata is publicly accessible on NCBI; however, detailed annotation data (e.g., gene coordinates, functional annotations) have not yet been released. This limitation makes chromosomal localization analyses unfeasible at this stage. (2) Motif-Conservation Analysis. This study described structural features of key neuropeptide precursors (e.g., AKH, proctolin, SIFamide, ecdysis-triggering hormone [ETH], FMRFamide, allatostatin-B, CAPA, PBAN, and tachykinin [TK]), including: conserved signal peptide positioning, proteolytic cleavage site organization, consistent arrangement of repetitive mature peptide sequences. However, cross-species comparisons of motif conservation were not conducted in this study.
Comments 5: [Finally, the authors must eliminate lingering typos by a meticulous proofread, ensure all gene/protein names are uniformly italicized, and combine supplementary figure numbering and legend data so melting curves, amplification plots, and calibration data are easily referenced.]
Response 5: We sincerely thank the reviewer for their meticulous feedback, which has significantly improved the clarity and rigor of our manuscript. The manuscript has been thoroughly proofread to eliminate typos and standardize formatting (e.g., ensuring consistent italicization/non-italicization of gene/protein names). Supplementary figure legends have been consolidated to facilitate seamless cross-referencing (see Figure S3 for representative examples, including melting curves, amplification plots, and calibration data). All figures and their corresponding datasets are now explicitly annotated to enhance accessibility.
We greatly value the reviewer’s attention to detail and hope these revisions meet their expectations. Should any further refinements be required, we welcome additional guidance.